# Neuroprotective Potential of Chrysin: Mechanistic Insights and Therapeutic Potential for Neurological Disorders

**DOI:** 10.3390/molecules26216456

**Published:** 2021-10-26

**Authors:** Awanish Mishra, Pragya Shakti Mishra, Ritam Bandopadhyay, Navneet Khurana, Efthalia Angelopoulou, Yam Nath Paudel, Christina Piperi

**Affiliations:** 1Department of Pharmacology and Toxicology, National Institute of Pharmaceutical Education and Research (NIPER)—Guwahati, Changsari, Kamrup 781101, Assam, India; 2Department of Pharmacology, School of Pharmaceutical Sciences, Lovely Professional University, Phagwara 144411, Punjab, India; ritambanerjee02@gmail.com (R.B.); navneet.18252@lpu.co.in (N.K.); 3Department of Nuclear Medicine, Sanjay Gandhi Post Graduate Institute of Medical Sciences (SGPGIMS), Lucknow 226014, Uttar Pradesh, India; pragya.mishra640@gmail.com; 4Department of Biological Chemistry, Medical School, National and Kapodistrian University of Athens, 11527 Athens, Greece; angelthal@med.uoa.gr (E.A.); cpiperi@med.uoa.gr (C.P.); 5Neuropharmacology Research Strength, Jeffrey Cheah School of Medicine and Health Sciences, Monash University Malaysia, Bandar Sunway 47500, Selangor, Malaysia; yam.paudel@monash.edu

**Keywords:** chrysin, antioxidant, neuroprotective agents, neurological disorders, epilepsy, neurodegenerative diseases

## Abstract

Chrysin, a herbal bioactive molecule, exerts a plethora of pharmacological effects, including anti-oxidant, anti-inflammatory, neuroprotective, and anti-cancer. A growing body of evidence has highlighted the emerging role of chrysin in a variety of neurological disorders, including Alzheimer’s and Parkinson’s disease, epilepsy, multiple sclerosis, ischemic stroke, traumatic brain injury, and brain tumors. Based on the results of recent pre-clinical studies and evidence from studies in humans, this review is focused on the molecular mechanisms underlying the neuroprotective effects of chrysin in different neurological diseases. In addition, the potential challenges, and opportunities of chrysin’s inclusion in the neurotherapeutics repertoire are critically discussed.

## 1. Introduction

The Global Burden of Disease (GBD) study and the World Health Organization (WHO) resources report that the overall health effects of neurological disorders have been underestimated. With a constant increase in the global age of the population, the augmented global burden of neurological disorders is posing a significant challenge to the maintenance of health care systems in developing and developed countries. Limited information is available pertaining to the prevalence, incidence and disease burden related to neurological disorders in India [1]. Over the past decade, the augmented incidence of neurological disorders has adversely affected quality of life, with severe socioeconomic consequences. Due to the elevated contribution of injury-related and non-communicable neurological disorders, research has been focused on the development of suitable management strategies. The most common neurological disorders include stroke, epilepsy, Alzheimer’s disease (AD), Parkinson’s disease (PD), multiple sclerosis (MS), cerebral palsy, brain tumor, and traumatic brain injury (TBI), being responsible for major disabilities worldwide [1,2,3,4,5].

Most of the currently available pharmacotherapeutic approaches provide merely symptomatic relief. Moreover, drug-associated adverse effects often complicate management, and further worsen the quality of life of these patients. Therefore, in the last decade, most of the research work has been focused on finding suitable alternatives with better safety profiles. In this regard, plant-derived functional foods with a wide variety of therapeutic and safety properties have gained growing attention among researchers. In general, the secondary metabolites of plants, including alkaloids, flavonoids, saponins, terpenes, etc., carry therapeutic potential. Flavonoids are bioactive molecules, derived from various plants and animal sources. Thousands of flavonoids have been reported carrying a broad spectrum of health benefits [3,4,6].

In the human diet, flavonoids represent the biggest group of plant-derived polyphenolic substances. On average, dietary flavonoid consumption is about 50–800 mg/day. As has been already extensively reviewed elsewhere, flavonoids exert a beneficial role on health due to their anti-oxidant, anti-inflammatory, antiviral and anti-carcinogenic properties via several cellular signaling pathways [7]. Flavonoid-rich foods, such as green tea, cocoa, and blueberry, exert beneficial effects via the interactions of flavonoids with several molecular targets. For instance, epigallocatechin gallate (EGCG), sequestered in red wine, chocolate and green tea, has been demonstrated to inhibit Aβ-induced neuronal apoptosis and caspase activity, promoting the survival of neurons in the hippocampus [8]. In addition, a blackberry-supplemented diet, which is enriched in polyphenols, has been associated with improved motor and cognitive performance in aged rat models [9]. Among family members, chrysin appears as a promising natural flavonoid, exhibiting an array of neuroprotective effects by attenuating oxidative stress, neuroinflammation, and apoptosis [3,6,7]. Chrysin, also known as chrysinic acid, belongs to the class of flavones. It is mainly obtained from honey, propolis, fruits and vegetables, primarily from the plants Yerba Santa, *Pelargonium crispum*, *Passiflora incarnate,* marsh skullcap and *Oroxylem indicum.* It possesses various pharmacological properties, including anti-inflammatory, anti-tumor, anti-asthmatic, antihyperlipidemic, cardioprotective, neuroprotective and renoprotective [3,8].

Although there are several reviews on the roles of flavonoids in health and disease, herein, we mainly address the neuroprotective effects of chrysin, specifically in neurological disorders, based on the accumulating pre-clinical evidence, and discuss its emerging therapeutic potential as well limitations that need to be overcome for its effective clinical use.

## 2. Chemistry and Pharmacokinetics of Chrysin

Chrysin consists of two fused rings (A and C) attached with a phenyl ring (B) at the second position of the C ring. In addition, at positions 5 and 7 of ring A, a hydroxyl group is attached (Figure 1) [3]. Polyphenols are not absorbed easily, especially in the form of esters, glycosides, and polymers. Due to their low absorption and high rate of metabolism and elimination, they possess poor intrinsic activity. Polyphenols degrade into aglycones and various aromatic acids after their hydrolyzation by intestinal enzymes. Aglycones are cardiac glycosides, considered as the most potent glycosides. Naturally occurring flavonoids get metabolized by phase I and phase II reactions (conjugation with methylation, sulfation and glucuronidation) and are eliminated from the body.

To address the pharmacological benefits and bioavailability of chrysin, it is necessary to understand the role of efflux transporters and the fate of its metabolites. There are three main transporters for chrysin conjugates: (a) the multidrug resistance-associated protein (MRP2), (b) the breast cancer resistance protein (BCRP), and (c) the ATP binding cassette (ABC). MRP2, also known as ABCC2, is a withdrawal efflux transporter that delivers anions, including drug conjugates and conjugated bilirubin. It is mainly expressed in the liver, kidney, and placenta. The chrysin metabolites are transported in Caco-2 cells through MRP2 [10]. These conjugates may be hydrolyzed by sulfatases and glucuronides to chrysin after their efflux into the small intestine. Studies using Caco-2 cell lines have shown that chrysin possesses favorable membrane transport properties [10]. However, a large amount of unchanged chrysin in stool samples indicates its poor intestinal absorption. BCRP (also known as ABCG2), an important efflux transporter of the ABC family of proteins for phase 2 metabolites (chrysin conjugates), are situated at the apical membrane of enterocytes and hepatocytes.

The anion inhibitor, MK–571, has been reported to reduce the elimination of chrysin metabolites (glucuronide and sulphate conjugates) in Caco-2 cells, suggesting that MRP2 may inhibit the efflux of chrysin glucuronide and sulfate conjugates up to 71% [11]. The lethal dose of chrysin through the oral route is 4350 mg/kg [12].

The major limitation of chrysin is its poor bioavailability, mainly due to its high metabolism. It is extensively metabolized by the intestine, liver, and several target cells, via conjugation, biotransformation, and the production of glucuronides and sulfate derivatives. Chrysin displays a very low distribution volume, and its oral bioavailability is about 0.003–0.02%. The urine and plasma levels of chrysin metabolites—sulfonate and glucuronide—are very low, while bile contains the highest concentrations [13]. However, significant efforts are currently being made towards overcoming this limitation, and are discussed below.

## 3. Potential Neuroprotective Mechanisms of Chrysin

Chrysin has been reported to exert neuroprotective effects through different mechanisms, including anti-oxidant, anti-inflammatory and anti-apoptotic functions, MAO inhibition and GABA mimetic properties. The neuroprotective mechanisms of chrysin are illustrated in Figure 2 and Figure 3.

### 3.1. Chrysin as an Anti-Oxidant Agent

Chrysin is a flavonoid, possessing a diphenylpropane (C6C3C6) skeleton system. In the structure–activity relationship studies, it has been shown that the diphenylpropane (C6C3C6) skeleton and the position of hydroxyl (-OH) substituents are very important for chrysin’s anti-oxidant and anti-inflammatory activities (Figure 1). The further substitution of these hydroxyl groups with methoxy or ethoxy groups causes reductions in the anti-oxidant and anti-inflammatory activities of chrysin, while C=C (between positions 2 and 3) is also important for these activities. The important pharmacophores of chrysin and the corresponding biological activities are illustrated in Figure 1 [14].

The Nuclear factor erythroid 2-related factor 2 (Nrf2), an important transcription factor for mediating the anti-oxidant effects, is upregulated by chrysin [15]. Upon activation, Nrf2 uncouples from Keap1 and migrates to the nucleus, where it binds to the anti-oxidant response element (ARE) and activates the downstream processing of heme oxygenase-1 (HO-1) and NAD(P)H quinone oxidoreductase 1 (NQO-1) (Figure 3). The downstream signaling of Nrf2 stimulates the production of anti-oxidant factors (SOD, GSH, and GST), and thus prevents oxidative stress-induced cellular damage [15].

The principal mediators of oxidative stress involve various types of reactive oxygen species (ROS) [16]. During oxidative stress, the fine balance between ROS production and removal is disrupted, leading to the accumulation of ROS inside the cell [16,17]. The increased ROS expression inside the cell leads to neurodegeneration via increased lipid peroxidation, mitochondrial dysfunction, and the activation of apoptotic cell death [18]. Chrysin exerts its neuroprotective effect mainly by reducing the prooxidants levels (ROS and lipid peroxidation) and augmenting the antioxidant defense factors (Figure 3) [19,20,21,22,23].

Chrysin can also indirectly affect the oxidative stress inside the cell by inducing the expression of various key antioxidant enzymes (Figure 2), including the superoxide dismutase (SOD), catalase (CAT) and glutathione peroxidase (GPx) [22,24,25,26]. Amongst the three isoforms of SOD present in our body, SOD1 and SOD3 play the most important roles in the antioxidant defense mechanism [27,28]. SOD, CAT and GPx exhibit antioxidant functions by catalyzing the dismutation of highly reactive superoxide to the less reactive hydrogen peroxide (H_2_O_2_), generating water and a dioxygen molecule from H_2_O_2_, while inhibiting lipid peroxidation [25,28,29,30]. Glutathione (GSH), a tripeptide abundant in the cytosol and cell organelles, is oxidized to glutathione disulfide (GSSG), and the rapid interconversion of GSH–GSSG–GSH maintains the cellular redox balance [31]. Chrysin has been shown to induce the expression of GSH, thus reducing oxidative stress [32].

### 3.2. Chrysin as an Anti-Inflammatory Agent

Inflammation is the body’s natural response to injury, infection, or trauma. Upon induction, it leads to a cascade of reactions that ultimately remove invading pathogens and start the wound-healing process along with angiogenesis [33]. The first key step in neuroinflammation is the activation of microglia cells [34]. The activation of microglia occurs mainly through the activation of the c-Jun N-terminal kinase (JNK) and the nuclear factor kappa light chain-enhancer of activated B cells (NF-κB) signaling pathway [35]. NF-κB signaling may be aggravated by the pathogen-associated molecular pattern (PAMP)/damage-associated molecular pattern (DAMP) molecules though Toll-like receptors (TLR) and the receptor for advanced glycated end products (RAGE) [36]. The downstream effector pathways of NF-κB include iNOS, cyclooxygenase-2 (COX-2) and various pro-inflammatory cytokines [37,38]. The increased levels of pro-inflammatory cytokines (IL-1β, TNF-α, and prostaglandins) have been shown to damage the blood–brain barrier (BBB) and induce apoptosis in neuronal cells [39,40]. The detection of DAMPs and PAMPs by pattern recognition receptors (PRRs) such as NLRP induces the formation of protein aggregates, finally leading to the formation of inflamosomes (Figure 2) [41]. Inflamosomes further induce the secretion of pro-inflammatory cytokines, such as IL-1β, IL-18 and pyroptosis, leading to a severe inflammatory cell response [42,43].

The anti-inflammatory activity of chrysin was shown to serve neuroprotectively after cerebral ischemia via modulation of estrogen receptors [44]. Chrysin has been also reported to modulate JNK and NF-κB expression and thus limit the progression of neuroinflammation [35,45,46,47]. It has been demonstrated to modulate the expression of NF-κB via the PI3K/AKT/mTOR and NLRP3 pathways, and to abrogate neuroinflammation [45,48,49,50]. Chrysin can also inhibit neuroinflammation via the mitigation of inflamosome formation by attenuating the NLRP3 signaling pathway [48,49]. On the other hand, chrysin directly mitigates pro-inflammatory cytokines (IL-1β, TNF-α, and prostaglandins) and exerts neuroprotective effects [46,47,51].

### 3.3. Chrysin as an Anti-Apoptotic Agent

Apoptosis is a natural cell death process, mediated by two mechanisms, i.e., the extrinsic and intrinsic pathways (Elmore, 2007). The extrinsic pathway starts after the binding of ligands, such as Fas and TNF-α, to the death receptors FasR and TNFR, respectively (Figure 2) [52]. This induces death signals inside the cell, leading to the activation of caspase-8, which further induces the activation of caspase-3. In turn, caspase-3 starts an array of reactions, known as the executional pathway, which ultimately leads to apoptosis [52,53,54,55]. On the other hand, the intrinsic pathway starts with the action of various external stimuli, including radiation, toxins, hypoxia, and oxidative stress, on the cell. These increase mitochondrial membrane permeability and the release of pro-apoptotic proteins into the cytosol [56,57]. These events further lead to caspase-9 activation, which ultimately activates caspase-3, thus activating the execution pathway [52,58]. Chrysin treatment was shown to reduce neurodegeneration by inhibiting the extrinsic, intrinsic and executionary pathways of apoptosis, and attenuating the expression of TNF-α, caspase-3/8 and oxidative stress (Figure 2) [47,59,60,61].

The BCL2 protein family plays a critical role in the control and regulation of the apoptotic process. During apoptosis, the expression of anti-apoptotic proteins (Bcl-2, Bcl-x, Bcl-XL, Bcl-XS, Bcl-w) is decreased and the expression of pro-apoptotic proteins (Bcl-10, Bax, Bak, Bid, Bad, Bim, Bik, and Blk) is increased [52]. It had been observed that chrysin administration is accompanied by the increased expression of anti-apoptotic proteins and a decreased expression of pro-apoptotic proteins [59,60,61,62,63]. Chrysin acting on the dysregulated stage of apoptosis-related proteins was shown to inhibit the neuronal death of cerebellar granular neurons in mice [61].

### 3.4. Chrysin as a MAO Inhibitor

Dopamine (DA) is one of the most important neurotransmitters present in the brain. Reduced DA levels are observed in the striatum and hippocampus brain regions of PD patients. The biosynthesis of dopamine starts with tyrosine hydroxylation by tyrosine hydroxylase, and via several reactions, it gets synthesized and stored inside the synaptic vesicles of dopaminergic neurons [64]. Chrysin was shown to reduce dopamine depletion and protect against the neurodegeneration of dopaminergic neurons of the brain [61,65]. It was also observed that chrysin treatment can significantly induce dopamine level recovery in the hippocampus and the prefrontal cortex region of the brain [66].

After secretion from the neurons, dopamine gets metabolized by MAO and COMT (into DOPAC and HVA) enzymes, and to some extent by alcohol/aldehyde dehydrogenase [64]. Chrysin inhibits MAO-A and MAO-B activity [61,67] and maintains dopamine levels in the brain. Additionally, its administration protects against changes in dopamine, DOPAC, and HVA levels in the brain of PD-induced animals [65,68], supporting the therapeutic potential of chrysin in PD.

### 3.5. Neuroprotective Role of Chrysin via a GABA Mimetic Action

GABA is the most important inhibitory neurotransmitter in the brain. It exhibits a neuroprotective effect by inhibiting brain injury, neuronal damage, autophagy (via the upregulation of Bcl-2/Bax ratio and activating AKT, GSK-3β and ERK signaling molecules) as well as neuronal cell death [69]. In this way, GABA demonstrates a therapeutic potential in various neurological disorders [69,70]. Most flavonoids, including chrysin, exert a GABA mimetic effect [71]. Chrysin has been shown to modulate the GABAA receptor and thus abrogate anxiety and depression-like behavior [72,73,74]. Therefore, it can act neuroprotectively via the modulation of GABAergic innervation.

## 4. Role of Chrysin in Different Neurological Disorders

The neuroprotective potential of chrysin has been widely explored in several neurological disorders. A comprehensive list of studies supporting the role of chrysin in different neurological disorders is provided in Table 1.

### 4.1. Chrysin in AD

AD is one of the most common progressive neurodegenerative disorders, characterized by dementia, while the oligomerization of amyloid-beta (Aβ) and the hyperphosphorylation of tau protein are considered as important pathological hallmarks. These abnormal protein aggregates initiate a variety of cellular responses (neuroinflammation, mitochondrial dysfunction, epigenetic alterations, and BBB changes), and eventually lead to neuronal death [108].

Studies have shown that chrysin may exert beneficial effects in AD disease models. The treatment of animals with chrysin-loaded magnetic PEGylated silica nanospheres has attenuated Aβ-induced memory impairment, possibly through the reduction of hippocampal lipid peroxidation levels and the elevation of antioxidant molecules (GSH, GPX, catalase, SOD, GSH), enabling neuroprotection [22,83,109]. In another study, free chrysin as well as CN-SLN were demonstrated to reverse learning impairment, along with a reduction in the neuroinflammation induced by Aβ, by lowering the expressions of IL-1β, IL-10 and TNF-α in the brain [82]. In MTZ-induced hypothyroid and associated dementia, chrysin treatment was demonstrated to reverse memory loss by overturning the decreased glutamate level and Na^+^/K^+^-ATPase activity [110].

### 4.2. Chrysin in PD

PD is the second most common neurodegenerative disorder, characterized by motor (bradykinesia, rigidity, tremors) and non-motor manifestations (pain, bladder and bowel disorders, depression). The chronic condition often disables the patient with shuffling gait, improper balance, and cognitive impairment [5,81].

Chrysin was shown to exhibit beneficial effects in various experimental models of PD. In the 1-methyl-4-phenyl-1, 2, 3, 6-tetrahydropyridine (MPTP)-induced experimental model of PD, chrysin treatment reduced the loss of dopaminergic neurons, possibly by mitigating apoptosis via the modulation of the AKT/GSK3β pathway and by restoring the imbalance in BCL2 family proteins [61]. Chrysin treatment has also caused a reduction in 6-hydroxydopamine (6-OHDA)-induced dopaminergic neuronal loss in substantia nigra pars compacta dopaminergic neurons, by mitigating oxidative stress through the activation of the NRF2/HO-1 pathway and neuroinflammation [65,78]. Chrysin restored striatal dopaminergic neuronal loss and improved the dopamine turnover in the striatum [77], supporting the protective effect of chrysin on motor functions [76].

### 4.3. Chrysin in Epilepsy

Epilepsy is a devastating neurological disorder characterized by unprovoked recurrent seizures, which might be attributed to aberrant neuronal activity. The pathomechanism of epilepsy is not yet fully understood. However, the imbalance in excitatory and inhibitory neurotransmission in the brain possibly contributes to the generation and propagation of seizures. In addition, alterations in the ion channels’ expression in the brain are considered as a plausible underlying cause [111,112,113].

The hydroethanolic extract of *Passiflora incarnata* L., its aqueous form (PIAE), as well as the hydroethanolic (PIHE) extract of *Passiflora incarnata* contain chrysin as an active ingredient. Their administration was shown to reduce pentylenetetrazol (PTZ)-induced seizure onset time, along with the severity and immobility period [86,87]. The administration of the ethanolic extract of *Pyrus pashia* fruits (containing chrysin as an active ingredient) exhibited anticonvulsant effects in PTZ-induced convulsions, along with antioxidant effects [85].

### 4.4. Chrysin in MS

MS is a relatively common disease of the central nervous system, characterized by inflammatory demyelination. The myelin sheath is essential for the protection of neuronal axons in the brain and the spinal cord, and MS is considered as an autoimmune disease. The animal model used for mimicking MS pathogenesis and the study of therapeutic interventions is the experimental autoimmune encephalomyelitis (EAE) model. The administration of chrysin in MS animal disease models was shown to improve clinical scores. Moreover, histone deacetylase inhibitors (HDACi) have been proposed as potential effective agents in neuroinflammatory diseases, including MS, due to their neuroprotective and immunosuppressive effects. Chrysin can block HDAC expression and reduce neuroinflammation in an EAE model [114]. It also causes weight loss, lowering cytotoxicity in animals, suggesting that HDAC inhibition by chrysin may be beneficial in the rodent EAE model [93].

Chrysin may also have significant effects on human DCs (dendritic cells). It can further eliminate the monocytes in peripheral blood mononuclear cells (PBMCs) in vitro and inhibit inflammatory cytokine production, along with the metabolic activity of PBMCs stimulated by lipopolysaccharide (LPS). Chrysin was further shown to induce phenotypic and functional changes in DCs [94]. Collectively, these findings suggest that chrysin-treated m-DCs may have the potential to reduce HLA-DR costimulatory molecules and induce T cell proliferation. Therefore, it has been proposed that the inhibitory effects of chrysin on antigen presentation may play a vital role in the pathogenesis of EAE and MS [109]. Additionally, chrysin has been reported to inhibit vascular cell adhesion molecule-1 expression via the inhibition of NF-κB/MAPK signaling, which is also significantly implicated in MS pathogenesis [46].

### 4.5. Chrysin in Traumatic and Ischemic Brain Injury

TBI is considered one of the common etiologies of neurological disorders. There are various clinical features of TBI, including reduced alertness, attention, memory loss, vison impairment, muscle weakness, etc. Treatment with chrysin was shown to reduce TBI-induced oculomotor dysfunction and memory impairment by inhibiting neuroinflammation and apoptosis via the upregulation of the Bcl-2 family and the downregulation of the Bax protein [62,89]. In another study, chrysin supported the alleviation of TBI-related anxiety and depression-like behavior. Furthermore, treatment with chrysin (10 and 20 mg/kg) was demonstrated to reduce brain edema after ischemic stroke [89]. Chrysin further reduced post-ischemic injury by alleviating the expression of pro-inflammatory cytokines (TNF-α and IL-10), as well as reducing pro-apoptotic (Bax) and augmenting anti-apoptotic (Bcl2) protein expression, thus exerting neuroprotective effects [45,89].

### 4.6. Chrysin in Gliomas

Gliomas are the most common brain tumors caused by the aberrant proliferation of glial cells, occurring both in the brain and the spinal cord. Glial cells, including astrocytes, oligodendrocytes, and microglia, support neuronal function. It has been shown that compounds found in propolis, such as CAPE, and chrysin may inhibit the NF-κB signaling pathway, a key signaling axis in glioma development and progression [115]. Moreover, it has been observed that the ethanolic extract of propolis interacts with the TMZ complex and may inhibit glioblastoma progression [115].

Chrysin treatment arrests the glioma cell cycle in G1 phase by increasing P21^(waf1/cip1)^ protein and activating P38-MAPK [100]. Chrysin combined with pine-needle extracts may regulate O-6-Methylguanine-DNA Methyltransferase (MGMT) suppression and AKT signaling, which play key roles in gliomagenesis [99]. Chrysin exhibited greater anti-glioblastoma activity compared to other compounds (PWE, pinocembrin, tiliroside) in GBM8901 cells. It was associated with reduced growth in the range of 25 to 100µM in a time-dependent manner in GBM8901 cells [99]. However, in contrast to other compounds, chrysin did not cause damage to other glial cell lines (detroit551, NIH3T3, EOC13.31 and rat mixed glial cells), suggesting that it may potentially display specific anti-glioblastoma properties without affecting normal cells [99]. The cleavage of caspase-3 and poly (ADP-Ribose) polymerase (PARP) was further detected upon chrysin treatment, and it was shown to reduce proliferation and induce apoptosis at high concentrations [98].

### 4.7. Possible Limitations of Chrysin and Strategies to Mitigate

Preclinical evidence supports the neuroprotective role of chrysin; however, clinical studies are limited due to the poor bioavailability of the compound [116,117]. The low bioavailability (less than 1%) is mainly attributed to its poor aqueous solubility, as well as its extensive pre-systemic and first pass metabolism [118,119]. The major portion of administered chrysin remains unabsorbed and is excreted in feces, providing evidence of its poor bioavailability [118,120,121,122]. Therefore, various approaches to improving the bioavailability of chrysin should be prioritized. Chemically, the basic scaffold of chrysin could be altered to gain better bioavailability and metabolic stability, retaining its neuroprotective mechanisms. The formulation-based approaches for the enhancement of brain bioavailability appear suitable, while retaining its neuroprotective mechanisms.

In recent years, studies have been focused on the development of several formulations to improve the efficacy of chrysin by overcoming the low bioavailability issue. Nanoformulation approaches have improved brain bioavailability. Chrysin-loaded sodium oleate-based nano emulsions were shown to inhibit the first pass glucuronide conjugation of chrysin, and led to a 4-fold increase in the peak plasma concentration [119]. Chrysin-loaded PLGA-PEG nanoparticle formulation enhanced the cellular uptake of chrysin in T47D and MCF7 cell lines [123]. Furthermore, co-crystals of chrysin were developed with cytosine and thiamine hydrochloride to enhance the dissolution and solubility rates by 3-4-fold, and thus, chrysin absorption was detected to be enhanced in in vivo and in vitro studies [124]. The development of chrysin-loaded solid lipid nanoparticles resulted in improved oral bioavailability and similar neuroprotective effects at lower doses [125]. Recently, the development of chrysin-loaded biotin-conjugated nanostructured lipid carriers (NLCs) successfully enhanced the peak plasma concentration of chrysin by 5–8-fold [126]. Overall, the nanoformulation approach has improved bioavailability and metabolic stability, while retaining the neuroprotective effect. Further, the suitability of this approach for the improvement of chrysin’s bioavailability is yet to be established in a clinical setup.

### 4.8. Conclusions and Future Perspectives

Emerging pre-clinical evidence has suggested that flavonoids present a promising backbone for future drug development related to the management of neurological diseases. Chrysin has emerged as an effective flavonoid and has gained extensive research attention. The neuroprotective effect of chrysin has been demonstrated through its anti-oxidant, anti-inflammatory, anti-apoptotic, and MAO inhibitory potential. Despite the several pre-clinical studies highlighting the plausible role of chrysin in various neurological disorders, clinical evidence is currently lacking, mainly due to its poor bioavailability and metabolic stability. The development of synthetic analogues of chrysin and nanoformulations may be promising strategies to overcome the pharmacokinetic challenges associated with chrysin. The further development of specific brain-targeting nanoformulations and the intranasal delivery of chrysin may have additional advantages in improving brain bioavailability, bypassing the first pass effect, and building the foundations for future clinical investigations.

## Figures and Tables

**Figure 1 molecules-26-06456-f001:**
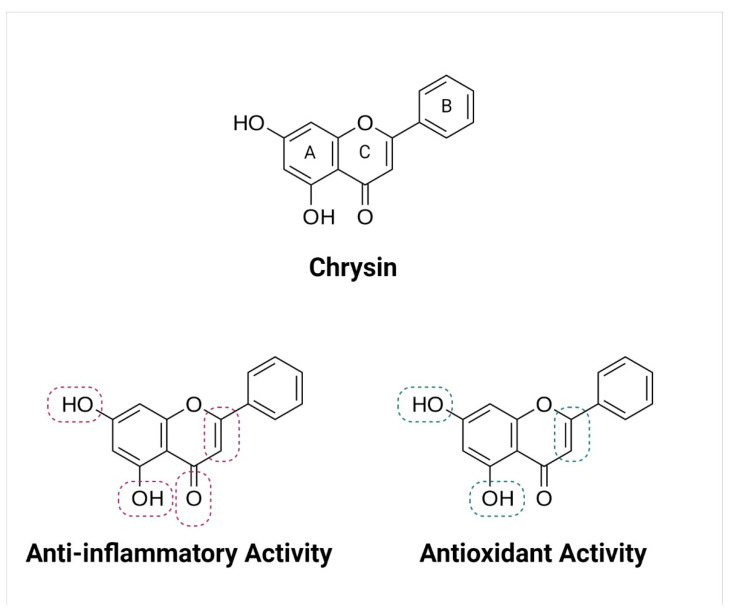
Chemical structure of chrysin and important pharmacophores for anti-inflammatory and anti-oxidant activity.

**Figure 2 molecules-26-06456-f002:**
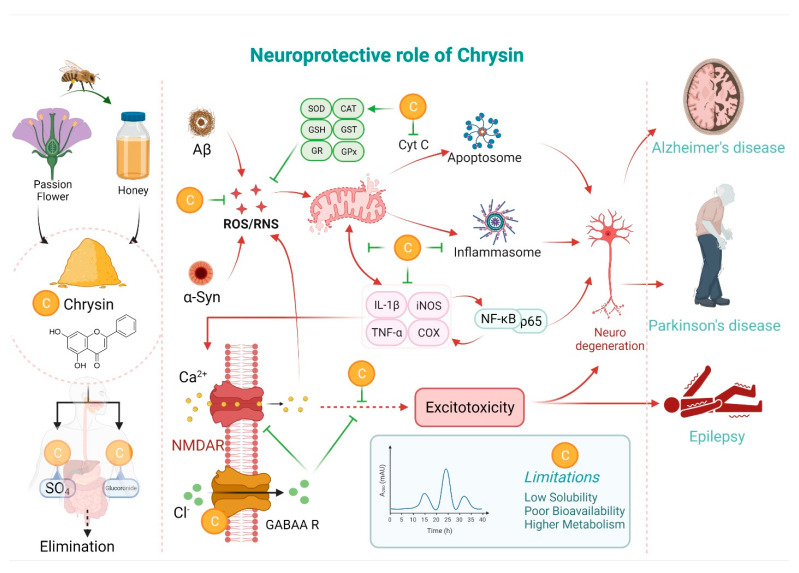
Effects of chrysin in the signaling networks associated with multiple neuropathological conditions.

**Figure 3 molecules-26-06456-f003:**
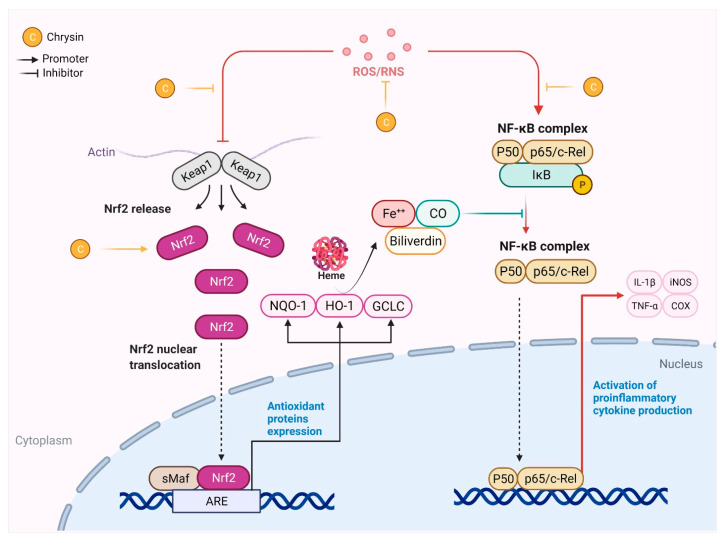
Modulation of the NRF-2 and NF-κB pathway by chrysin. ROS/RNS mediate the alteration of NRF-2 signaling and interconnect with the NF-κB signaling pathway. NRF-2 signaling activates the expression of antoxidant proteins viz, heme oxygenase-1 (HO-1), NAD(P)H quinone oxidoreductase 1 (NQO-1) and Glutamate-cysteine ligase catalytic subunit (GCLC, a rate-limiting enzyme for glutathione synthesis). The degradation of heme produces carbon monoxide (CO), which suppresses redox-sensitive NF-κB activation. The plausible sites of action of chrysin are illustrated.

**Table 1 molecules-26-06456-t001:** Experimental evidence supporting the neuroprotective role of chrysin in various neurological disorders.

#	Dose and Route	Experimental Model	Animal/Cell Lines	Outcome	Ref.
**In Parkinson’s Disease**					
1.	Chrysin (100 µM)	2,4-dinitrophenol-induced mitophagy	*Caenorhabditis elegans* (*Bristol N2 wild-type, ucp-4 deletion mutant, pdr-1, and zdIs5*)	Chrysin served as mitochondrial uncoupler and mitigated neurodegeneration possibly via PINK1/Parkin mitophagy	[75]
2.	Chrysin (50, 100, and 200 mg/kg, p.o.) for 5 days	MPTP-induced rodent model of PD	Male C57BL/6 J mice (18–22 g)	Improvement in motor dysfunction and dopaminergic neuroprotection in nigro-striatal region, possibly by mitigating oxidative stress and neuroinflammation	[68]
3.	Chrysin (10 mg/kg, i.g.) for 28 days	6-OHDA-induced rodent model of PD	Female C57B/6 J mice (30–40 g, 20 months)	Improvement in motor and cognitive functions along with reduction in oxidative stress, neuroinflammation	[65]
4.	Chrysin (50 mg/Kg, i.p.) for 4 weeks	Rotenone-induced rodent model of PD	Sprague–Dawley rats	Improvement in motor impairments and attenuation of nigrostriatal dopaminergic neurodegeneration	[76]
5.	Chrysin (10 mg/kg, p.o.) for 28 days	6-OHDA-induced rodent model of PD	Male C57B/6 J mice (20–30g, 90 days)	Improvement in motor functions and restoration of dopaminergic neurons, inflammatory cytokines, and neurotrophic factors levels	[77]
6	Chrysin (50 mM)	MPP^+^-induced neurotoxicity	Primary Cerebellar Granule Neuron Culture	Neuroprotection via inhibition of apoptosis by activating MEF2D via AKT-GSK3β signaling	[61]
7	Chrysin (10, 100 mg/kg, p.o.)	MPTP-induced rodent model of PD	Male C57BL/6 mice (28 ± 2 g, 8–9 weeks)	Restoration of dopaminergic neuronal loss via anti-apoptotic, activation of the AKT-GSK3β/MEF2D pathway, and inhibition of MAO-B activity	[61]
8	Chrysin (25 µM)	6-OHDA-induced neurotoxicity	Rat pheochromocytoma (PC12) cells	Neuroprotection by attenuating oxidative stress (NRF-2/HO-1 pathway), neuroinflammation (NF-κB/iNOS pathway)	[78]
9	Chrysin (3, 6, and 12 µM)	6-OHDA-induced dopaminergic neurotoxicity	Zebrafish larvae (AB strain)	Protection of dopaminergic neurons	[78]
10	Chrysin (40 µM)	MPP^+^-induced neurotoxicity	Primary mesencephalic neurons	Neuroprotection of mesencephalic dopaminergic neurons via attenuation of oxidative stress and apoptosis	[79]
**In Alzheimer’s Disease**					
1	Chrysin (50 µM)	Amylin-induced amylin amyloidosis	C6 Rat Glioma Cell lines	Reduction in amylin amyloidosis	[80]
2	Chrysin (10 mg/kg, p.o.) for 3 months	Zinc-induced cognitive impairment and amyloidosis	Male Swiss mice	Improvement in cognitive functions and neuroprotection of hippocampal neurons	[81]
3	Chrysin loaded lipid-core nano capsules (1 and 5 mg/kg., p.o.) for 14 days	Aβ_1–42_-induced animal model of AD	Female Swiss mice (30–35 g, 18–22 months)	Improvement in learning and memory impairment via attenuation of oxidative stress and neuroinflammation	[82]
4	Chrysin loaded magnetic PEGylated silica nanospheres (30 µM)	Aβ-induced amyloidosis in hippocampal culture model	Sprague–Dawley rat (neonates)	Improved antioxidant profile and protection against Aβ-induced oxidative stress	[83]
5	Chrysin (50 mg/kg) for 21 days	Acrylamide or gamma-irradiation induced neurotoxicity	Male Wister rats (120–150 g)	Neuroprotective effect via attenuation of oxidative stress, amyloidosis, and apoptosis	[84]
6	Chrysin loaded sloid lipid nanoparticles (5, 10, 50, 100 mg/kg, p.o.) for 21 days	Aβ_25–35_-induced rodent model of AD	Male Sprague–Dawley rats (250–300 g)	Improved memory impairment and amelioration of hippocampal neuronal loss possibly via mitigation of neuronal loss	[22]
**In Epilepsy**					
1	Chrysin-loaded PLGA nanoparticle (5 and 10 µg/mL)	PTZ-induced kindling	Wistar rats	Anticonvulsant effect through mitigating oxidative stress via the NRF2/HO-1 pathway	[15]
2	Chrysin (2.5, 5, and 10 mg/kg; p.o.)	PTZ-induced convulsions	Male Charles Foster rats (180–220 g)	Anticonvulsant effect possibly via alleviation of oxidative stress in hippocampus and cortex	[85]
3	Hydroethanolic extract of *Passiflora incarnata* (150, 300, and 600 mg/kg, i.p.) for 11 days	PTZ-induced kindling	Male Swiss mice (20–30 g)	Presence of chrysin in the extract and anticonvulsant, antidepressant effects	[86]
4	Extract of *Passiflora incarnata* (150, 300, and 600 mg/kg; i.p.)	PTZ-induced convulsions	Male Swiss mice (25–35 g)	Presence of chrysin in the extracts and anticonvulsant effect of the extracts	[87]
5	Chrysin (40 µg, i.c.v.)	PTZ-induced convulsions	Swiss mice (22–28 g)	Anticonvulsant effect via activation of benzodiazepine receptors	[88]
**Chrysin in Traumatic Brain Injury**					
1	Chrysin (25, 50 and 100 mg/kg, p.o.)	Closed head weight-drop-induced rodent model TBI	Male Wistar rats (250–300 g)	Improved neurobehavioral impairments possibly via modulation of inflammation and apoptosis	[89]
2	Chrysin (25, 50 and 100 mg/kg, p.o.)	Closed head weight-drop-induced rodent model TBI	Adult male Wistar rats (250–300 g)	Improved motor coordination and memory impairment possibly via anti-oxidant and anti-apoptotic effects	[62]
**Chrysin in Ischemic Brain Injury**					
1	Chrysin (10, and 20 mg/kg) for 7 days	Middle cerebral artery occlusion-induced cerebral ischemia/reperfusion injury model	Male Sprague-Dawley rats (250–280 g)	Reduction in ischemia/reperfusion injury in brain, possibly via alleviation of proinflammatory cytokine release and improvement of antioxidant defense by activating the PI3K/Akt/mTOR pathway	[45]
2	Chrysin (10, 30, and 100 mg/kg, p.o.) for 21 days	Bilateral common carotid arteries occlusion model of cerebral ischemia reperfusion injury	Male Wistar rats (250–300 g, 6 month)	Improvement in cognitive impairment and restoration of hippocampal neurons possibly by reducing oxidative stress and PGE2 levels	[90]
3	Chrysin (10, 30, and 100 mg/kg, p.o.) for 21 days	Bilateral common carotid arteries occlusion model of cerebral ischemia reperfusion injury	Male Wistar rats (250–300 g)	Improvement in cognitive impairment possibly by alleviating neuroinflammation	[91]
4	Chrysin (30 mg/kg, p.o.) for 14 days	Bilateral common carotid arteries occlusion model of cerebral ischemia reperfusion injury	Male Wistar rats (200–250 g)	Neuroprotection against ischemia reperfusion injury possibly by attenuating oxidative stress	[92]
5	Chrysin (50 mg/kg, p.o.) for 10 days	Bilateral common carotid arteries occlusion model of cerebral ischemia reperfusion injury	Male C57BL/6 J mice (18–22 g)	Reduction in degenerative changes in neurons possibly by mitigating oxidative stress	[93]
6	Chrysin (75 mg/kg, p.o.) for 7 days	Middle cerebral artery occlusion-induced ischemia reperfusion injury	Male C57/BL6 mice (10–12 weeks)	Reduction in neurological deficit scores and infarct volumes, possibly via inhibition of neuroinflammation (by suppression of NF-κB, COX-2, and iNOS expression)	[94]
7	Chrysin (30, and 100 mg/kg; i.g.) for 26 days	Bilateral common carotid arteries occlusion model of cerebral ischemia reperfusion injury	Male Wistar rats, (330–350 g)	Improvement in dementia and neurodegeneration possibly via attenuation of oxidative stress and neuroinflammation	[95]
**Chrysin in gliomas**					
1	Chrysin (5, 30, 60, 120, and 240 µM)	Human glioblastoma cell lines	T98, U251, U87 cells	Anticancer activity in glioblastoma cell lines possibly via the ERK/Nrf2 signaling pathway	[96]
2	Chrysin (10, 20, 40, 80 and 120 µM) and Cisplatin (0.5, 1.0 and 2.0 µM) combination	Human glioma cell lines	U87 cells	Potentiation of antiproliferative effect of Cisplatin	[97]
3	Chrysin (50 µM)	Human glioblastoma cell lines	GL-15 and U251 cells	Damaged mitochondria, and rough endoplasmic reticulum, apoptosis, and reduction in MMP-2 expression	[98]
4	Chrysin (100 µM)	Human glioblastoma cell lines	GBM8901 glioblastoma cells	Induction of apoptosis and suppression of migration and invasion. Inhibition of temozolomide-induced autophagy and O6-methylguanine-DNA	[99]
5	Chrysin (10, 30, and 50 µM)	Rat glioma cell line	C6 glioma cells	Induction of G1 phase cell cycle arrest through induction of p21^Waf1/Cip1^ and inhibition of proteasome activity	[100]
**Chrysin in MS**					
1	Chrysin (20 mg/kg, i.g.) for 25 days	Myelin oligodendrocyte glycoprotein-induced EAE	Male C57BL/6 mice (20–25 g)	Reduction in HDAC activity, GSK-3β and proinflammatory cytokine release	[101]
2	Chrysin (100 mg/kg, i.g.) for 3 days	Myelin oligodendrocyte glycoprotein-induced EAE	Female C57BL/6 mice (6–8 weeks)	Amelioration of EAE and anti-inflammatory and immune suppressive effects via suppression of dendritic cells and Th1 cells	[102]
**Miscellaneous**					
1	Chrysin (0.05 mM)	Diclofenac-induced neurotoxicity	SH-SY5Y cells	Neuroprotective effect of mitigating oxidative stress and apoptosis	[60]
2	Chrysin (400 µM)	Cyclophosphamide-induced neurotoxicity	SH-SY5Y cells	Neuroprotection via suppression of oxidative stress and apoptotic cell death	[59]
3	Chrysin (5, and 10 μM)	LPS-induced inflammation	BV2 microglia cells and Primary mouse microglia cells	Suppression of neuroinflammation by downregulating NF-κB/TRAF6 pathway and upregulating zinc figure protein A20	[103]
	Chrysin (25 and 50 mg/kg, p.o.) for 4 days	LPS-induced neuroinflammation	Male Balb/c mice (20 ± 2 g, 8 weeks)		
4	Chrysin (20 mg/kg, i.g.) for 28 days	Methimazole induced hypothyroidism and associated neurobehavioral impairments	Female C57BL/6 mice (3–4 months)	Improvement in depression-like behavior via improvement of cortical and hippocampal serotonin levels and hippocampal dopamine levels	[66]
5	Chrysin (60, 80, 100, 150 and 200 µg/mL)	LPS-induced neuroinflammation	RAW264.7 macrophage cells	Reduction in inflammatory response by blocking the JAK-STAT pathway mediated by ROS	[51]
6	Chrysin (10–100 μM)	LPS-induced in vitro study	Mouse cerebral vascular endothelial (bEnd.3) cells	Reduction in VCAM-1 expression by inhibition of NF-κB/MAPK pathway resulted in anti-inflammatory effect	[46]
7	Chrysin (100 mg/kg, p.o.)	Ammonium chloride-induced neuroinflammation	Male Wistar rats	Attenuation of neuroinflammation by reducing expression of pro-inflammatory markers (TNF-α, IL-1β, IL-6, NF-κB, iNOS and COX-2) in the brain	[104]
8	Chrysin (50 mg/kg, p.o.) for 14 days	3-nitro propionic acid-induced neurotoxicity	Male Wistar rats (250–300 g)	Improvement in neurobehavioral impairments, mitochondrial dysfunction, oxidative stress, and apoptosis	[63]
9	Chrysin (10 mg/kg, p.o.) for 60 days	Age-related memory decline in mice	Male Swiss Mice (3 and 20 months)	Improvement in age-related memory decline by attenuating oxidative stress and Na^+^/K^+^ ATPase activity in prefrontal cortex and hippocampus	[26]
10	Chrysin (5 and 20 mg/kg, p.o.) for 28 days	Chronic unpredictable mild stress	Female C57B/6 J mice (20–25 g, 90 days)	Alleviation of depression-like symptoms possibly by upregulation of BDNF, NGF levels, antioxidant defense factor (GPx, GR, Catalase) and reduction in ROS level and Na^+^/K^+^ ATPase activity	[105]
11	Chrysin (30 and 100 mg/kg, i.g.) for 26 days	Weight-drop method-induced spinal cord injury model	Wistar rats (230–250 g)	Augmentation in neuronal recovery and reduction in pro-inflammatory markers and iNOS expression	[106]
12	Chrysin in vitro (0.5–5 µM) for 12 and 24 h	Acrylamide-induced neurotoxicity	PC12 cells	Neuroprotection	[107]
13	Chrysin in vivo (12.5, 25, and 50 mg/kg)	Acrylamide-induced toxicity in vivo	Male Wistar rats (230–250 g)	Reduction in gait abnormality	[107]

## Data Availability

No new data were created or analyzed in this study. Data sharing is not applicable to this article.

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
