# Peer review of "Neuroprotective Potential of Chrysin: Mechanistic Insights and Therapeutic Potential for Neurological Disorders"

_molecules, 2021, doi:10.3390/molecules26216456_

Round 1

Reviewer 1 Report

The review written by Mishra et al., described the neurological potential of chrysin a dietary flavonoid. Even do the manuscript is well written and is well presented with tables and figures. The topic has extensive reports and several of those reports are not cited here.  For instance, a report on the same topic was published recently see PMID:34328092. Additionally, to several other reports: PMID: 31472146; PMID: 31785348; PMID: 34198618, etc

Personally, I do not think that recurrence of the same topic is relevant. So, the authors must state really well what is different of this review in comparison to the other (above). In other words, what is new in this review?  To be consider to Molecules. otherwise, it is not suitable for publication.

Author Response

Author’s response: We would like to thank the reviewer for his/her comments and the valuable time for reviewing this manuscript. Although several manuscripts are available on a similar topic, the novelty of this manuscript relies on the elaboration of preclinical investigations, highlighting the limitations of chrysin and proposing strategies to overcome them as well as emphasize future directions. In addition, we have discussed the neuroprotective effect of chrysin across several neurological disorders (AD, PD, TBI, MS, Epilepsy, and Ischemic stroke to mention a few) in a comprehensive way in a single manuscript which will ease the readers. Moreover, we have considered all reviewer's suggestions/comments and we have significantly revised the manuscript, increasing its quality.

Reviewer 2 Report

Figure 2 is confusing and requires more explanation, this figure should be reference more throughout the document when these different possible mechanisms are discussed within each section. Or more figures should be included within each section more clearly showing the reader the pathways being discussed and altered by Chrysin’s activity.

In section 3.1 the authors discuss NRF2 activation yet that isn’t shown anywhere in Fig 2,  Why not?

Page 3, section 3.1 paragraph 1, line 5, typo substitution is written as “substation”

Mention some about SAR of flavones but give no graphic or discuss further, this could be expanded upon.

Authors mention the limited bioavailability of chrysin yet have no discussion about how this could be improved or how to best overcome the potential PKPD challenges posed by this flavone scaffold. Expand the conclusions section to actually discuss and analyze the literature. This article is a good start and has reviewed the studies but there is no synthesis of new knowledge or critical evaluation of how to move this molecule forward. The authors have mentioned major issues with this compound for future development and yet present the findings of these studies without any analysis on which studies may have been impacted by the simple anti-oxidant activity in an in vitro system and when these studies may actually represent the possibility to translate these findings into clinical application.

There was some discussion about SAR but no discussion or mention about how flavones typically have very flat SAR and are not amenable to further optimization or synthetic modification. The authors could present supporting evidence for why they believe that chrysin is a good candidate for modification and what potential areas for optimization exist which would help it to overcome its many issues (solubility, metabolism, non-specific effects).

Overall this article needs some general grammatic editing for clarity but more importantly the authors need to not only present what previous studies found but discuss and add new knowledge to the field and space.

Author Response

Reviewer 2

Comment 1: Figure 2 is confusing and requires more explanation, this figure should be reference more throughout the document when these different possible mechanisms are discussed within each section. Or more figures should be included within each section more clearly showing the reader the pathways being discussed and altered by Chrysin’s activity.

Author’s response: In the revised manuscript, Figure 2 is referenced more in the text and more figures have been added.

Comment 2: In section 3.1 the authors discuss NRF2 activation yet that isn’t shown anywhere in Fig 2, Why not?

Author’s response: As suggested Figure 3 has now been added to the manuscript which clearly indicates the NRF-2 and NF κB pathways being modulated by chrysin.

Comment 3: Page 3, section 3.1 paragraph 1, line 5, typo substitution is written as “substation”

Author’s response: The typographical mistake has been corrected in the manuscript.

Comment 4: Mention some about SAR of flavones but give no graphic or discuss further, this could be expanded upon.

Author’s response: Figure 1 has now been changed to include the chemical structure of chrysin and SAR of chrysin. The SAR has also been elaborated in section 3.1.

Comment 5: Authors mention the limited bioavailability of chrysin yet have no discussion about how this could be improved or how to best overcome the potential PKPD challenges posed by this flavone scaffold. Expand the conclusions section to discuss and analyse the literature.

Author’s response: A new section (5. Possible Limitations of Chrysin and strategies to mitigate) has been added to the manuscript. Also, the conclusion part has been expanded, to analyse the literature and provide authors insight on the clinical application of chrysin.

Comment 6: This article is a good start and has reviewed the studies but there is no synthesis of new knowledge or critical evaluation of how to move this molecule forward. The authors have mentioned major issues with this compound for future development and yet present the findings of these studies without any analysis on which studies may have been impacted by the simple antioxidant activity in an in vitro system and when these studies may actually represent the possibility to translate these findings into clinical application.

Author’s response: We have clearly described the possible limitations of chrysin and the strategies to counteract the challenges considering the existing preclinical evidence for their better translational aspects for clinical applications. These changes have been incorporated in section 5. 

Comment 7: There was some discussion about SAR but no discussion or mention about how flavones typically have very flat SAR and are not amenable to further optimization or synthetic modification. The authors could present supporting evidence for why they believe that chrysin is a good candidate for modification and what potential areas for optimization exist which would help it to overcome its many issues (solubility, metabolism, non-specific effects).

Author’s response: We have now discussed the possible pharmacokinetic challenges, possible approaches to overcome the solubility, metabolism, and bioavailability issues of chrysin and highlighted the nanoformulation based approaches. Based on the evidence we proposed that chrysin is a good candidate if its brain bioavailability is improved.

Overall, this article needs some general grammatic editing for clarity but more importantly the authors need to not only present what previous studies found but discuss and add new knowledge to the field and space.

Author’s response: We have extensively revised the grammar and editing mistakes throughout the manuscript, and we have given emphasis to the future scope of Chrysin in neurotherapeutics.

Reviewer 3 Report

The manuscript entitled: “Neuroprotective Potential of Chrysin: Mechanistic Insights and Therapeutic Potential for Neurological Disorders” is very interesting, but needs major corrections.

Major problems:

  1. In the abstract, the Authors wrote: "Based on the recent pre-clinical and clinical evidence, this review is focused on the molecular mechanisms underlying the neuroprotective effects of chrysin in different neurological diseases." The study focuses on in vitro and pre-clinical research. While I understand that clinical trials are not described because they are not available, it is suggested in the abstract that RCTs will also be analyzed. This leaves a certain dissatisfaction.
  2. There is no mention of chrysin permeability through the BBB. Due to the low bioavailability of chrysin, I propose to describe a short paragraph on new methods of its delivery, e.g. nanocapsulation.
  3. While I fully agree with the statement that chrysin increases the activity of antioxidant enzymes via the Nrf2/HO-1, is this the only signaling pathway involved in antioxidant protection?
  4. The descriptions of chrysin's action (section 3.1-3.5) provide overly detailed general descriptions of e.g. inflammation, apoptosis. In my opinion, the main emphasis should be on the action of chrysin. I propose to make one subsection describing the neurorestorative effect of chrysin, in which the antioxidant, anti-inflammatory, anti-apoptotic, MAO inhibitory, and GABA mimetic effects will be described. Especially since they are interconnected through signaling pathways.
  5. In the section "Role of Chrysin in different Neurological Disorders", the research is insufficiently described. It is not clear what the research model was used? What neurological outcomes have been caused by the modulation of individual pathways?
  6. Table 1 should be rearranged and shortened. It would be much clearer if it summarizes the specific chrysin action, signaling pathways, and possibly potential doses, e.g. disease / neurorestorative effect / signaling pathway / dose (optional) / bibliography. Currently the table is too long and unreadable.
  7. There is no discussion of the research results presented in the paper. Although the molecular mechanism has been described quite accurately, it does not provide a reference to the clinical aspect.

Minor comments:

  1. Unfortunate statements, e.g. "bran cancer" rather "brain tumor"
  2. Not all abbreviations have been explained
  3. "Chrysin is highly bound to plasma proteins (> 99%) and has been reported safe at dose ranging from 300-625 [12,13]." - There is no given unit

Author Response

Reviewer 3

The manuscript entitled: “Neuroprotective Potential of Chrysin: Mechanistic Insights and Therapeutic Potential for Neurological Disorders” is very interesting, but needs major corrections.

Major problems:

  1. In the abstract, the Authors wrote: "Based on the recent pre-clinical and clinical evidence, this review is focused on the molecular mechanisms underlying the neuroprotective effects of chrysin in different neurological diseases." The study focuses on in vitro and pre-clinical research. While I understand that clinical trials are not described because they are not available, it is suggested in the abstract that RCTs will also be analysed. This leaves a certain dissatisfaction.

Author’s response: We would like to thank reviewer for their critical observation. The suggested change has now been included in the abstract section.

  1. There is no mention of chrysin permeability through the BBB. Due to the low bioavailability of chrysin, I propose to describe a short paragraph on new methods of its delivery, e.g. nanocapsulation.

Author’s response: A separate section of approaches to mitigate the low bioavailability of chrysin has been added in the manuscript.

  1. While I fully agree with the statement that chrysin increases the activity of antioxidant enzymes via the Nrf2/HO-1, is this the only signaling pathway involved in antioxidant protection?

Author’s response:  Authors agree with the reviewer’s assertion. Therefore, the sentence has been revised for better clarity.

  1. The descriptions of chrysin's action (section 3.1-3.5) provide overly detailed general descriptions of e.g. inflammation, apoptosis. In my opinion, the main emphasis should be on the action of chrysin. I propose to make one subsection describing the neurorestorative effect of chrysin, in which the antioxidant, anti-inflammatory, anti-apoptotic, MAO inhibitory, and GABA mimetic effects will be described. Especially since they are interconnected through signaling pathways.

Author’s response: As suggested, subsections of section has been dissolved to summarize the neurorestorative mechanism of chrysin (without having any additional subsection).

  1. In the section "Role of Chrysin in different Neurological Disorders", the research is insufficiently described. It is not clear what the research model was used? What neurological outcomes have been caused by the modulation of individual pathways?

Author’s response: The detailed description of experimental model in various neurological disorders and their neuroprotective mechanisms have been summarized in Table (1). Therefore, repetition of data has been avoided in the section 4. If reviewer prefers to add details in the text, we will happily include them.

  1. Table 1 should be rearranged and shortened. It would be much clearer if it summarizes the specific chrysin action, signaling pathways, and possibly potential doses, e.g. disease / neurorestorative effect / signaling pathway / dose (optional) / bibliography. Currently the table is too long and unreadable.

Author’s response: We would like to thank the reviewer for his critical evaluation. The Table 1 has been prepared according to the sequence of data described in the main text, i.e., we have divided the preclinical findings according to the diseases affected. In this way, after reading the text, a reader can easily follow and understand the table. If the reviewer prefers a correction in the table formatting, we will be happy to perform it. 

  1. There is no discussion of the research results presented in the paper. Although the molecular mechanism has been described quite accurately, it does not provide a reference to the clinical aspect.

Author’s response: Further data on the clinical aspects on the neurotherapeutics use of this molecule is currently missing in the literature. 

Minor comments:

  1. Unfortunate statements, e.g. "bran cancer" rather "brain tumor"

Author’s response: The error has now been corrected.

  1. Not all abbreviations have been explained

Author’s response: Due care has been taken to enlist all the abbreviations in the manuscript.

  1. "Chrysin is highly bound to plasma proteins (> 99%) and has been reported safe at dose ranging from 300-625 [12,13]." - There is no given unit.

Author’s response: The sentence has now been clarified and elaborated now.

Round 2

Reviewer 1 Report

Still the manuscript is not to the level of Molecules, Chrysin the molecule of choice is poorly bioavailable and the authors just mentioned it and not touch how this fact will be tackle. Not mentioned. Another important point is chrysin is a dietary flavonoid maybe relate the common consume and the biological activity which other molecules will be interesting. In general reviews about a single molecule are not attractive to the public.

Author Response

RESPONSE TO REVIEWERS’ COMMENTS

Reference#: molecules-1347932 Submission Title: Neuroprotective Potential of Chrysin: Mechanistic Insights and Therapeutic Potential for Neurological Disorders. Mishra et al., Molecules, 2021.

Reviewer 1

Still the manuscript is not to the level of Molecules, Chrysin the molecule of choice is poorly bioavailable and the authors just mentioned it and not touch how this fact will be tackle. Not mentioned. Another important point is chrysin is a dietary flavonoid maybe relate the common consume and the biological activity which other molecules will be interesting. In general reviews about a single molecule are not attractive to the public.

Author’s Response: We would like to thank the Reviewer for the points raised.

In order to highlight and discuss in more detail the poor bioavailability of chrysin we added the following paragraph in Section 2 “The major limitation of chrysin is its poor bioavailability mainly due to its high metabolism. It is extensively metabolized by intestinal, liver and several target cells, via conjugation, biotransformation and production of glucuronides and sulfate derivatives. Chrysin displays very low distribution volume and its oral bioavailability is about 0.003–0.02%. Urine and plasma levels of chrysin metabolites – sulphonate and glucuronide- are very low, while the bile contains the highest concentrations [13]. However, significant attempts are currently made towards overcoming this limitation, and are discussed below.”

We have more extensively discussed the ways to tackle this limitation in Section “5. Possible Limitations of Chrysin and strategies to mitigate”, where we mention recent strategies aiming to improve the bioavailability of chrysin via nanoformulation approaches, including chrysin-loaded sodium oleate-based nano emulsions, chrysin- loaded PLGA-PEG nanoparticle formulation, co-crystals of chrysin developed with cytosine and thiamine hydrochloride, chrysin-loaded solid lipid nanoparticles and chrysin-loaded biotin-conjugated nanostructured lipid carriers (NLCs) formulations.

In order to note the important bioavailability issue, we have also mentioned this limitation in the Conclusion Section “Development of synthetic analogues of chrysin and nanoforulation may appear as promising strategies to overcome the pharmacokinetic challenges associated with chrysin. Further development of specific brain-targeting nanoformulation and intranasal delivery of chrysin may have added advantages on improved brain bioavailability and bypassing first pass effect and would be helpful for making basic grounds for future clinical investigations. “

To briefly relate the effects of chrysin with other flavonoids of the common human diet, we added the following Paragraph in the Introduction Section:

“In the human diet, flavonoids represent the biggest group of plant-derived polyphenolic substances. On average, the dietary flavonoid consumption is about 50-800 mg/day. As it has been already extensively reviewed elsewhere, flavonoids exert a beneficial role on health due to their anti-oxidant, anti-inflammatory, antiviral and anti-carcinogenic properties via several cellular signaling pathways [7]. Foods rich in flavonoids such as green tea, cocoa, and blue berry exert beneficial effects via the interactions of flavonoids with several molecular targets. For instance, epigallocatechin gallate (EGCG), sequestered in red wine, chocolate and green tea, has been demonstrated to inhibit Aβ-induced neuronal apoptosis and caspase activity, promoting the survival of neurons in the hippocampus [8] In addition, blackberry-supplemented diet, which is enriched in polyphenols, has been associated with improved motor and cognitive performance in aged rat models [9].”

In addition, we added the following Paragraph at the end of the Introduction Section:

“Although there are several reviews on the role of flavonoids on health and disease, herein, mainly based on the accumulating preclinical evidence, we focus on the neuroprotective effects of chrysin specifically in neurological disorders, discussing its emerging therapeutic potential and limitations that need to be overcome for its effective clinical use.”

Reviewer 2 Report

The authors have addressed the major concerns with the previous version. While I still feel that there could be more scholarship and analysis in this review, the authors have added considerbale efforts and discussion to this paper improving its possible impact for those in the field. 

There are still some minor editing issues for the journal to address but the science and description of previous studies appears wel done and complete. 

Some additional clarification on word choice regarding "clinical" usage and evidence in humans may be worth adding as noted by another reviewer. 

Author Response

RESPONSE TO REVIEWERS’ COMMENTS

Reference#: molecules-1347932 Submission Title: Neuroprotective Potential of Chrysin: Mechanistic Insights and Therapeutic Potential for Neurological Disorders. Mishra et al., Molecules, 2021.

Reviewer 2

The authors have addressed the major concerns with the previous version. While I still feel that there could be more scholarship and analysis in this review, the authors have added considerable efforts and discussion to this paper improving its possible impact for those in the field. There are still some minor editing issues for the journal to address but the science and description of previous studies appears well done and complete. Some additional clarification on word choice regarding "clinical" usage and evidence in humans may be worth adding as noted by another reviewer.

Author’s response: We would like to thank the Reviewer for the positive feedback. As suggested, we altered this sentence in the Abstract as follows: “Based on the results of recent pre-clinical studies and clinical evidence evidence from studies in humans,”.

We also corrected several remaining editing issues in the text.

Reviewer 3 Report

Thank you for considering my comments. I only still feel unsatisfied with the abstract, because "clinical evidence" relates directly to clinical trials, and "pre-clinical studies" to in vitro and animal models studies. Therefore, I propose to delete "clinical" in the abstract so as not to mislead the reader.

Author Response

RESPONSE TO REVIEWERS’ COMMENTS

Reference#: molecules-1347932 Submission Title: Neuroprotective Potential of Chrysin: Mechanistic Insights and Therapeutic Potential for Neurological Disorders. Mishra et al., Molecules, 2021.

Reviewer 3

Thank you for considering my comments. I only still feel unsatisfied with the abstract, because "clinical evidence" relates directly to clinical trials, and "pre-clinical studies" to in vitro and animal models studies. Therefore, I propose to delete "clinical" in the abstract so as not to mislead the reader.

Author’s response: We thank the Reviewer for this important mention. As also suggested by Reviewer 2, we deleted the word “clinical” in order to avoid misleading, and we altered the referred sentence in the Abstract as follows: “Based on the results of recent pre-clinical studies and clinical evidence evidence from studies in humans,”.
